# Polygenic Risk of Hypertriglyceridemia Is Modified by BMI

**DOI:** 10.3390/ijms23179837

**Published:** 2022-08-30

**Authors:** Virginia Esteve-Luque, Marta Fanlo-Maresma, Ariadna Padró-Miquel, Emili Corbella, Maite Rivas-Regaira, Xavier Pintó, Beatriz Candás-Estébanez

**Affiliations:** 1Cardiovascular Risk Unit, Internal Medicine Department, Hospital Universitari de Bellvitge, L’Hospitalet de Llobregat, 08907 Barcelona, Spain; 2Bellvitge Biomedical Research Institute (IDIBELL), L’Hospitalet de Llobregat, 08908 Barcelona, Spain; 3CIBEROBN Fisiopatología de la Obesidad y Nutrición, Instituto de Salud Carlos III, 28029 Madrid, Spain; 4Clinical Laboratory, Hospital Universitari de Bellvitge, L’Hospitalet de Llobregat, 08907 Barcelona, Spain; 5Centro de Salud San Juan, 31011 Pamplona, Spain; 6Faculty of Medicine, Universitat de Barcelona UB, L’Hospitalet de Llobregat, 08007 Barcelona, Spain; 7Clinical Biochemistry Laboratory, SCIAS-Hospital de Barcelona, 08034 Barcelona, Spain

**Keywords:** polygenic hypertriglyceridemia, dyslipidemia, hypercholesterolemia, obesity, genetic risk score, gene–environment interaction

## Abstract

**Background**: Genetic risk scores (GRSs) have partially improved the understanding of the etiology of moderate hypertriglyceridemia (HTG), which until recently was mainly assessed by secondary predisposing causes. The main objective of this study was to assess whether this variability is due to the interaction between clinical variables and GRS. **Methods**: We analyzed 276 patients with suspected polygenic HTG. An unweighted GRS was developed with the following variants: c.724C > G (*ZPR1* gene), c.56C > G (*APOA5* gene), c.1337T > C (*GCKR* gene), g.19986711A > G (*LPL* gene), c.107 + 1647T > C (*BAZ1B* gene) and g.125478730A > T (*TRIB* gene). Interactions between the GRS and clinical variables (body mass index (BMI), diabetes mellitus, diet, physical activity, alcohol consumption, age and gender) were evaluated. **Results**: The GRS was associated with triglyceride (TG) concentrations. There was a significant interaction between BMI and GRS, with the intensity of the relationship between the number of alleles and the TG concentration being greater in individuals with a higher BMI. **Conclusions**: GRS is associated with plasma TG concentrations and is markedly influenced by BMI. This finding could improve the stratification of patients with a high genetic risk for HTG who could benefit from more intensive healthcare interventions.

## 1. Introduction

Hypertriglyceridemia (HTG) is a frequent disorder in the general population and is related to the risk of cardiovascular disease and pancreatitis [1]. Multiple genetic variants associated with HTG have been found in genome-wide association studies (GWASs) [2]. Individually, these variants have minimal effects on triglyceride (TG) concentrations and provide only limited information for clinically assessing individual risk [3]. However, the association of different genetic variants related to TG metabolism, each with a relatively small effect, can be expressed by a genetic risk score (GRS) and can explain a significant proportion of the variability of plasma TG concentrations and predict the risk of developing HTG [4,5,6,7,8,9].

Nonetheless, the accumulation of these allelic variants can only explain 10–20% of the susceptibility to these disorders, and most of the genetic susceptibility in HTG patients remains to be established. Other factors such as epigenetics, gene–gene and gene–environment [10] interactions and omnigenics [11,12] may contribute to explaining this variability or heritability.

An unbalanced diet, sedentary lifestyle, obesity, metabolic syndrome and type II diabetes are secondary causes of HTG [13,14], and their prevalence has progressively increased worldwide in recent decades. Synergistic effects between GRS and clinical variables related to secondary HTG may help explain some of the variability of these traits [15,16].

Interactions between genetic risk factors and clinical phenotypes may account for some of the unexplained heritability of plasma lipid traits. In the present study, we calculated one GRS based on single nucleotide polymorphisms (SNPs) related to TG (all identified through a large-scale GWAS) and clinical variables including diet, physical activity and body mass index (BMI) in a group of patients followed in our hospital. We studied the influence of clinical variables and the relationship between each GRS and the blood lipid profile based on the hypothesis that these clinical variables modify the polygenic risk associated with high TG levels.

## 2. Results

Table 1 shows the baseline characteristics of the study participants. The study sample was divided according to the GRS values (GRS ≤6 or >6). Both groups were comparable, and between them, only differences in TG, CT or non-HDL-c were found. The mean age of the study sample was 52.1 (standard deviation (SD): 10.5) years, and 73.9% were males. The mean BMI was 28.7 kg/m^2^, and 61.2% were current or former smokers, 34.1% had hypertension and 17.8% had DM. The mean score on the diet questionnaire was 7.5. The mean of the highest TG levels recorded from the clinical records of the patients was 7.84 (95% confidence interval (CI): 7.0 to 8.6) mmol/L. The mean TC, non-HDL-c and HDL-c values corresponding to the sample with the highest TG concentrations were 6.71 (95% CI 6.5 to 6.9) mmol/L, 5.74 (95% CI 5.5 to 6.0) mmol/L and 0.98 (95% CI 0.94 to 1.01) mmol/L, respectively.

Patients with DM exhibited good metabolic control as they presented a mean glycated hemoglobin (HbA1c) of 6.8% (SD of 0.7), DM (results not shown in Table 1).

Appendix A shows the baseline characteristics of the study participants separated by gender. No differences in TG concentrations were found among different genders; only differences in waist circumference, alcohol consumption, smoking habit, TC and HDL-c were found.

The GRS ranged from 4 to 11. Patients from this sample showed a higher prevalence of allelic variants than that expected in the general population (Appendix A).

The GRS explained 7% (R^2^ calculated by a simple regression model) of the variability in TG concentrations (results not shown).

A gradual and direct relationship between GRS and TG concentrations was observed (Figure 1), being more notable in patients with a higher GRS. Thus, an increase in a risk allele represented an increase of 1.3 to 1.59 mmol/L in TG concentrations in individuals with genetic loads greater than six SNPs, while in subjects with genetic loads below six SNPs, the increase in TG concentrations was practically zero for each increase in one allele in GRS.

There was a significant interaction between BMI and GRS, with no evidence of such interactions with gender, age, diet, DM, physical activity or alcohol consumption (Appendix A).

In Table 2, the results of the multiple linear regression model developed to predict TG concentrations based on the GRS, BMI (their interaction), DM and diet are presented. The table shows the interaction between GRS and BMI; here, interaction signifies that the effect of GRS on TG values depends on BMI values, as presented in Figure 2.

Table 2 presents the contribution of the other variables to the TG variation independently of GRS.

Diet quality showed a constant and inverse relationship with TG levels, with an improvement of one point in the diet questionnaire corresponding to a decrease of 0.23 mmol/L in TG levels. This relationship was independent of the GRS score. On average, TG levels were 2 mmol/L higher in DM patients than in non-DM patients (Table 2).

Figure 2 was created to represent the effect of this interaction, showing a prediction for the worst triglyceride concentrations for three BMI values. To represent the different groups (normal weight, overweight and obesity), the BMI value located in the median position for each group was selected. A BMI of 23.5 kg/m^2^ was selected to represent the normal-weight group, 27 kg/m^2^ for the overweight group and, lastly, 32 kg/m^2^ for the obesity group. To carry out the statistical analysis, the variables Diet = 7 points and DM = No were kept constant, and the calculation was conducted for each GRS and BMI value described. The greater the BMI, the greater the slope of the association between the GRS and TG concentrations. Each increase in one risk allele in the GRS was associated with an increase of 0.53 mmol/L in triglyceride concentrations in normal-weight patients, 1.08 mmol/L in overweight patients and 1.87 mmol/L in patients with obesity.

This figure was constructed by applying the regression equation corresponding to Table 2. The TG concentrations predicted were the result of modifying one BMI and GRS value, keeping Diet (=7 points) and DM (=No) unchanged.

Figure 2 also shows the increase in TG concentrations based on the change in the BMI group for each GRS group. This increase in the TG concentrations predicted for the variation in the BMI group was more evident for a GRS greater than 6. Thus, in patients with a GRS of 6, an increase in BMI from 23.5 to 27 represented an increase of 0.004 mmol/L in TG concentrations, and an increase in BMI from 27 to 32 represented an increase of 0.005 mmol/L in TG concentrations. Marked increases in TG concentrations were observed with a higher genetic load and higher BMI. The greatest increase in TG was, therefore, found in patients with a GRS of 10 and an increase in BMI from 27 to 32, which represents an increase in TG concentrations of 3.085 mmol/L.

## 3. Discussion

The main finding of this study is that the polygenic risk for HTG is attenuated in normal-weight subjects.

Blood TG levels are regulated by multiple genes [18,19], except in patients with familial chylomicronemia syndrome. Numerous SNPs associated with blood TG concentrations have been identified using GWAS candidate gene approaches [2]. However, it has been observed that, individually, these SNPs contribute very little to the variations in HTG levels [20]. In this sense, multiple GRSs were created to combine the relatively small additive effects of individual SNPs, and these have proven to be good predictors of plasma TG concentrations [4,5,6,7,8,9].

Similarly, in the present study, we developed a score related to TG concentrations. This relationship was not linear; having a higher genetic load (the number of alleles) implies a greater increase in TG levels. For genetic loads of greater than six SNPs in the GRS, a higher increase in TG concentrations was observed for each increment of one allele compared with those observed for genetic loads of less than six SNPs, which were almost zero. These results are similar to those already shown in that the association between GRS and TG levels was statistically stronger [9,21] when the risk score was higher [4,5,6,7,8,9].

Visceral obesity, overweight and global obesity [22,23], as well as GRS, have been related to HTG; however, few studies have analyzed the gene–environment [24] relationship in polygenic HTG, which could be responsible for this lack of heritability and the variability of action of these SNPs [25] in the phenotypic manifestation.

In this study, BMI status significantly influenced the effect of these genetic variants on plasma TG concentrations. Thus, obesity or overweight amplified the effect of GRS by further increasing TG levels, while normal weight attenuated this effect. In this regard, similar results have been described in studies evaluating the effect of abdominal obesity [15,24,25] or the degree of physical conditioning [16] on these risk scores, observing that they also acted as modifiers.

Polygenic HTG is the result of an imbalance between TG production (with fatty acids from the diet) and TG clearance (affected by the accumulation of SNPs) [10]. In this sense, the greater the accumulation of these SNPs, the lower the environmental factors that are required to manifest HTG and vice versa. As observed in the present study, the influence of BMI on TG concentrations was only observed in patients with a high genetic load.

The interaction between BMI and GRS can be explained by different factors. These genes encode proteins that alter both hepatic TG synthesis and peripheral lipolysis, and obesity is not only related to a greater contribution of fatty acids to the liver but also to impaired clearance of TG-rich lipoprotein particles. 

The *GCKR* gene product [26], glucokinase regulatory protein, regulates glucokinase (GCK) activity competitively with respect to the substrate glucose, inhibiting GCK activity. Hepatic GCK activity enhances glycolytic flux, promoting hepatic glucose metabolism and increasing malonyl CoA availability, a major substrate for de novo hepatic lipogenesis. The variant NM_003904.4:c.*724C>G of the *ZPR1* gene corresponds to an intergenic zone located near the *APOA5-A4-C3* gene cluster and was related to the development of clinical HTG by decreasing apolipoprotein A5 (apoA5) concentrations [27]. The *APOA5* [28,29] gene encodes for apoA5, a lipoprotein lipase (LPL) activator [30]. Both *ZPR1* and *APOA5* genetic variants are related to a decrease in the lipolytic function of lipoprotein lipase. *LPL* encodes lipoprotein lipase, a major determinant of peripheral lipolysis of TG-rich lipoproteins. 

In obese individuals, overnutrition is related to a greater contribution of fatty acids (FA) to the liver [31], a fundamental substrate for TG synthesis. In addition, an increase in carbohydrates is related to an increase in de novo lipogenesis (DNL), which is responsible for synthesizing TG from carbohydrates. Obesity is also related to insulin resistance, which causes the insulin-mediated inhibition of DNL to disappear. Obesity is, however, not only related to an increase in the contribution of FA to the liver for the production of TG but also affects the clearance of these particles [32] as it is associated with an increase in apolipoprotein CIII [33] levels, an LPL inhibitor. 

Our results suggest that reducing BMI may also lessen the effects of the GRS on TG levels. Other studies reinforce this concept by finding that lifestyle interventions appeared to partially mitigate the effect of genetic contribution [34,35]. Further research in larger populations is needed to know whether such interventions could be useful, especially across different ethnicities and different genetic risk profiles.

Some studies carried out in children reported that the GRS was a good predictor of HTG in adulthood [36]. Thus, the performance of these scores at early ages, with effects that are clearly modifiable by clinical variables, would allow undertaking actions related to possible cases of HTG much earlier by influencing dietary measures, weight or physical activity, thereby reducing the cases of manifest HTG and the associated cardiovascular disease.

Some studies suggest that older people respond less efficiently to lifestyle interventions [37], thus favoring the detection of these individuals with an elevated GRS at a very early age, even before they present HTG, in order to carry out lifestyle interventions to achieve the greatest possible effect.

Some limitations of our study merit consideration. First, our sample size was relatively small, which might have led to a type 2 error; however, the patients were duly selected by medical specialists. Second, during our literature search, we found differing numbers of candidate SNPs to create the GRS, and, finally, we selected those that showed the greatest effect on plasma TG concentrations and risk of HTG in a GWAS. However, the population presented a high prevalence of some of these SNPs, making it difficult to assess their effect on TG levels.

## 4. Materials and Methods

### 4.1. Population Selection

Unrelated patients (*n* = 276) with primary HTG, aged 18 to 80 years, visited in the Lipid and Vascular Risk Unit of our tertiary hospital were included in this study.

Patients with serum TG concentrations between 2 and 10 mmol/L in at least two blood tests were included. Patients with severe systemic or life-threatening disease, or with the following secondary causes of HTG, were excluded: chronic liver disease; stage 4–5 chronic kidney disease or on dialysis; type II diabetes mellitus with bad glycemic control (HbA1c >10%) or hypothyroidism (with thyroid-stimulating hormone >8 mU/L); alcohol abuse (defined as intake >40 g—4 standard drink units (SDUs)—a day in males and >20 g—2 SDUs—in females) [37]; and treatment with drugs that may cause secondary HTG (Appendix A). Dysbetalipoproteinemia was discarded by apolipoprotein E genotype analysis.

HbA1c >10% was chosen as this is the value at which insulinopenia should be considered.

### 4.2. Variables Selected

All patients provided information regarding their age, sex, profession, drug therapy, smoking habit and alcohol consumption. Cigarette smoking was categorized as “none”, “past” or “current” and alcohol consumption as the number of SDUs per week. Diet assessment was performed using a validated and standardized diet questionnaire [17], and physical activity was assessed by self-reported hours of physical activity per week. Anthropometric measures such as weight, height and waist were measured during the first visit, and BMI was calculated as weight/height^2^ (kg/m^2^). Data about hypertension, diabetes mellitus (DM) and hypercholesterolemia were also registered.

All blood tests were performed after eight hours of fasting, in outpatients a few days before each of the follow-up visits in our Vascular Risk Unit. During the follow-up visits, active diseases were ruled out. Fasting lipid profile results corresponding to the highest plasma TG concentrations and the respective drug treatments were recorded for each patient. The results of plasma TG analysis were collected from the clinical records of the patients via the institutional electronic information system. Fasting lipid profiles included plasma concentrations of total cholesterol (TC); low-density lipoprotein cholesterol (LDL-c), calculated using the Friedewald equation only when TG concentrations were lower than 2.3 mmol/L (200 mg/dL); high-density lipoprotein cholesterol (HDL-c); non-high-density lipoprotein cholesterol (non-HDL-c); TG; apolipoprotein B; and apolipoprotein A1. Serum concentrations of HbA1c were also recorded. 

LDL-c was calculated using the Friedewald equation. This was only used when TG concentrations were lower than 2.3 mmol/L (200 mg/dL); when TG concentrations were higher, only non-HDL-c was calculated.

A history of atherothrombotic cardiovascular disease, including coronary artery disease, cerebrovascular disease and peripheral arterial disease, was recorded through anamnesis and a review of medical records. Coronary artery disease was defined as acute myocardial infarction, stable or unstable angina, coronary artery bypass graft or percutaneous transluminal coronary angiography with evidence of significant atherosclerotic disease. Cerebrovascular disease included both past stroke and a transitory ischemic attack, and peripheral arterial disease was defined as the presence of a pathological ankle–brachial index (<1), or clinical manifestations of intermittent claudication.

A history of pancreatitis was also recorded from the clinical records of the patients via the institutional electronic information system, or through self-reports when the episode occurred in other hospitals.

### 4.3. Biochemical Analyses

All biochemical analyses were performed in plasma using a COBAS 800 automated analyzer (Roche Diagnostics^®^). HbA1c was measured in whole blood using the HA-AutoA1C 8180 autoanalyzer from Menarini^®^ (Florence, Italy).

CT: Reagent: Chol2 gen2 COBAS 501/502 for COBAS 8000. Test principle: colorimetric enzymatic method. Cholesterol esters are broken down by the action of cholesterolesterase in free cholesterol and fatty acids. Cholesterol oxidase catalyzes the oxidation of cholesterol in cholest 4 in 3-one and peroxide of hydrogen. In the presence of peroxidase, the hydrogen peroxide formed produces an oxidative linkage of phenol and 4-aminoantipyrine (4-AAP) to form a red quinonimine dye.

LDL-c: LDL-c was estimated using the Friedewald equation if TG concentrations were below 2.3 mmol/L. If not, non-HDL-c was calculated instead.

TG: Reagent: TRIGL. COBAS 501/502 for COBAS 8000. The present method is based on the work of Wahlefeld and uses a lipoprotein lipase derived from microorganisms to rapidly and completely hydrolyze TG to glycerol, with subsequent oxidation to dihydroxyacetonephosphate and hydrogen peroxide. The resulting hydrogen peroxide reacts under the catalytic action of peroxidase with 4-aminophenazone and 4-chlorophenol to form a red dye in a Trinder endpoint reaction. The color intensity of this red dye is directly proportional to the concentration of TG and can be measured photometrically.

HDL-c: Reagent: HDLC4 COBAS 501/502 for COBAS 8000. Test principle: homogeneous enzymatic colorimetric test. Non-HDL lipoproteins including LDL, VLDL and chylomicrons combine with polyanions and detergent to form a complex water soluble. In this complex, the enzymatic reaction of CHE and CHOD versus non-HDL lipoproteins occurs so that only the HDL particles can react with CHER and CHOD. The concentration of HDL-c is determined enzymatically by CHER and CHOD. Cholesterol esters are quantitatively degraded to free cholesterol and fatty acids by CHER.

### 4.4. Genetic Testing

The following allelic variants were selected according to the greatest effect on plasma TG concentrations and the risk of HTG shown in a GWAS [5]: *ZPR1* gene rs964184 (NM_003904.4:c.*724C>G), *APOA5* gene rs3135506 (NM_001166598.1:c.56G>C), *GCKR* gene rs1260326 (NM_001486.3:c.1337T>C), *LPL* gene rs12678919 (NC_000008.11:g.19986711A>G), *BAZ1B* gene rs7811265 (NM_032408.3: c.107+1647T>C) and *TRIB* gene rs2954029 (NC_000008.11: g.125478730A>T).

The characteristics of these genetic variants are shown in Appendix A. Analysis of these allelic variants showed a higher prevalence than expected in a GWAS in these patients (Appendix A).

DNA was extracted from peripheral blood using an automated DNA purification system (Maxwell^®^ RSC Instruments, Promega, Madison, WI, USA) and stored at −80 °C.

Genotyping was carried out using the TaqMan SNP Genotyping Assay (assay IDs: C_8907629_10, C_25638153_10, C_2862880_1, C_9639494_10, C_2632556_10 and C_15954645_10; Applied Biosystems^®^, Foster City, CA, USA) in 96-well plates including positive and negative controls. Real-time polymerase chain reaction (PCR) was carried out with the 7500 Real-time PCR System, Applied Biosystems (Thermo Fisher Scientific, Waltham, MA, USA), following standard recommendations. Briefly, 1 µL of Assay Mix was mixed with 10 µL Supermix SsoAdvanced (Biorad^®^, Hercules, CA, USA), 2 µL genomic DNA (20 ng/µL) and purified water up to 20 µL. The resulting mixture was heated to 50 °C for 2 min and 95 °C for 10 min in the thermal cycler. This was then followed by 40 cycles of denaturing at 95 °C for 15 s and annealing/extending at 60 °C for 1 min.

### 4.5. Selecting SNPs and Development of the Genetic Risk Score

An unweighted GRS was created in which each allele was assumed to impart the same effect on plasma TG concentrations. The genotypes of the six variants studied were encoded as “0” when the variant was non-present, “1” if one allele was present (heterozygous) and “2” if two alleles were present (homozygous). As six allelic variants were selected, the score could range between 0 and 12.

### 4.6. Statistical Analysis

Qualitative variables were described as absolute frequencies, quantitative variables were described as means with standard deviations (SDs) and non-normal variables were described as means and interquartile ranges.

A GRS was created by adding the number of mutated alleles present in each individual for each of the six allelic variants; for the statistical analysis, the three individuals with scores of 11 were grouped with individuals with scores of 10. The study population was divided into two groups depending on whether they presented GRS > 6 or GRS < 6. Both groups were comparable.

A bivariate analysis was carried out to evaluate the clinical variables related to the highest TG concentrations: qualitative variables were compared by analysis of variance (ANOVA) and quantitative variables with the Pearson correlation test (data not shown). Variables that were statistically significant together with age and sex (considered clinically relevant) were considered to be included in the multivariable regression model. Interactions between GRS and different clinical variables (BMI, gender, age, DM, diet, physical activity and alcohol consumption) were evaluated; only the interaction between BMI and GRS was significant. Finally, a multiple linear regression model with GRS and BMI (their interaction), including DM, diet, age and gender, was created. Using a backward stepwise procedure, the variables age and sex were excluded, as they did not contribute significant information to the model.

## 5. Conclusions

There was a positive association between the GRS and TG concentrations, and these scores may be useful for predicting HTG, even before it appears as a phenotypic trait. The main finding of our study is that the polygenetic contribution to HTG was significantly modified by BMI, with a higher BMI being associated with a stronger influence of the GRS on TG concentrations. This finding could improve the stratification of patients with a high genetic risk for HTG, who could benefit from more intensive healthcare interventions to lower their BMI even before the phenotypic trait is present.

## 6. Patents

This section is not mandatory but may be added if there are patents resulting from the work reported in this manuscript.

## Figures and Tables

**Figure 1 ijms-23-09837-f001:**
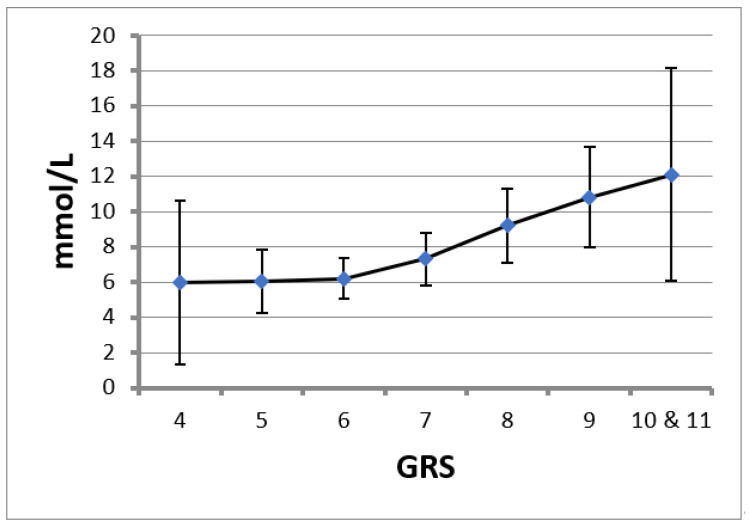
Triglyceride (TG) levels for each genetic risk score (GRS). Data are the mean and 95% confidence interval.

**Figure 2 ijms-23-09837-f002:**
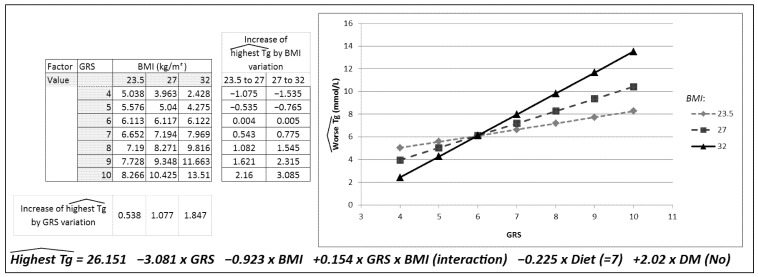
Descriptive model graphically representing the interaction between the genetic risk score (GRS) and body mass index (BMI).

**Table 1 ijms-23-09837-t001:** Baseline characteristics of the study participants.

CLINICAL VARIABLES	Total	GRS		*p*
	*n* = 276	≤6 *n* = 113	>6 *n* = 163	
**Sex (male) (%)**	204 (73.9%)	85 (75.2%)	119 (73%)	0.680
**Age (years)**	52.1 (10.5)	51.2 (9.5)	52.7 (11.1)	0.260
**BMI (kg/m^2^)**	28.7 (3.8)	28.3 (4.2)	28.9 (3.5)	0.265
**Waist circumference (cm)**	100.4 (12.5)	102 (15.9)	99.3 (9.8)	0.137
**Current or former smoker (%)**	169 (61.2%)	72 (63.7%)	97 (59.5%)	0.481
**Alcohol consumption (units/w)**	1.2 (2.7)	1.5 (3.4)	0.9 (2.0)	0.069
**Diet score (*)**	7.5 (3.3)	7.3 (3.4)	7.6 (3.3)	0.440
**Physical activity (hours/week)**	2.6 (3.0)	2.5 (3.2)	2.7 (2.9)	0.558
**HTA (%)**	94 (34.1%)	40 (35.4%)	54 (33.1%)	0.696
**DM (%)**	49 (17.8)	18 (15.9%)	31 (19%)	0.509
**Chronic renal disease (%)**	19 (6.9)	6 (5.3%)	13 (8%)	0.390
**Atherothrombotic cardiovascular disease**	75 (27.2%)	30 (26.5%)	45 (27.6%)	0.846
**Pancreatitis (%)**	2 (0.7%)	0 (0%)	2 (1.2%)	0.515
**Total cholesterol (mmol/L) ***	6.7 (1.8)	6.4 (1.8)	6.9 (1.8)	0.029
**HDL-c (mmol/L) ***	0.98 (0.32)	1.0 (0.33)	0.96 (0.31)	0.269
**LDL-c (mmol/L) ***	3.6 (1.5)	3.7 (1.6)	3.4 (1.4)	0.272
**non-HDL-c (mmol/L) ***	5.7 (1.8)	5.4 (1.8)	6.0 (1.8)	0.018
**TG (mmol/)**	5.3 (3.4 to 9.6)	4.6 (3.0 to 6.9)	5.8 (3.9 to 11.6)	<0.001
**ApoB (mg/dL)**	1.2 (0.4)	1.2 (0.4)	1.2 (0.4)	0.624
**HbA1c (%)**	5.87 (0.71)	5.88 (0.76)	5.85 (0.67)	0.764

Results are expressed as the mean and standard deviation (SD), median (interquartile range) or *n* (percentage). * Diet score was calculated according to a standardized questionnaire [17].ƒ GRS: genetic risk score; DM: diabetes mellitus; ApoB: apolipoprotein B; HbA1c: glycated hemoglobin; BMI: body mass index; HTA: hypertension; TC: total cholesterol; LDL-c: low-density lipoprotein cholesterol; HDL-c: high-density lipoprotein cholesterol; non-HDL-c: non-high-density lipoprotein cholesterol; TG: triglycerides. TC, HDL-c, LDL-c and non-HDL-c values correspond to the blood test with the highest TG values for each patient.

**Table 2 ijms-23-09837-t002:** Statistical regression model used to predict TG concentrations.

	Regression Coefficient (95% CI)	Standardized Coefficient	*p*	R^2^
Constant	26.151 (0.28 to 52.02)		0.048	0.131
GRS	−3.08 (−6.80 to 0.64)	−0.70	0.104	
BMI (kg/m^2^)	−0.923 (−1.83 to −0.20)	−0.53	0.045	
GRS * BMI interaction	0.15 (0.03 to 0.28)	0.07	0.020	
Diet	−0.23 (−0.46 to 0.01)	−0.11	0.056	
DM	2.02 (0.01 to 4.03)	0.12	0.049	

CI: confidence interval; GRS: genetic risk score; BMI: body mass index; Diet: diet score 1–14 points as explained in the Materials and Methods; DM: diabetes mellitus.

## Data Availability

Not applicable.

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
