# Peer review of "Polygenic Risk of Hypertriglyceridemia Is Modified by BMI"

_ijms, 2022, doi:10.3390/ijms23179837_

Round 1

Reviewer 1 Report

The findings were of interest while there were several parts to be reconsidered for improving the paper. The novelty of the study may be more stressed.

Refer to the following suggestions:

1.      Of important, the authors should do the gender-separated subanalysis because TG values are known to have a gender difference.

2.      The effects of genetic types of hypertriglyceridemia (i.e., Type I, III, V) on the results would be more discussed.

3.      In Methods, the authors could describe about the unrelated or related condition of each patient, for instance based on the history taking (or family history).

4.      In Methods, the authors could describe about the history of pancreatitis.

5.      In Methods, the authors could describe the reason for definition of the poor control of diabetes (9% of HbA1c).

6.      In Methods, the authors should describe the condition of sampling blood in each patient (e.g., concrete fasting, acute or chronic, non-stress frustration, etc.) because the triglyceride is sensitive to changing conditions among any lipid measures.

7.      In Methods, the authors should describe the drug names for exclude the patients with medications. This should be determined to make a strict study protocol.

8.      In Methods, the authors should describe the reagents and the measurement principles of each lipid. The CV of each measurement should be added to it.

9.      In Methods, the authors could describe the application of Friedwald equation to hypertriglyceridemia because the equation is not accurate in such high triglyceride values.

10.  In Analysis, the authors could state the valid theory to determine cutoff of GRS (6), which is universal.

11.  TG values sould not display the mean plus sd levels due to their wide deviations in Table 1.

12.  Many readers might not know the outcome for each variable in Table 2.

13. In Abstract, 1, 2, 3 and 4 in front of Background, Methods, Results and Conclusions had better be deleted.

14. In Abstract, TG should be abbreviated when triglyceride first appeared.

15. In Conclusions, the word ‘important association’ did not seem to be objective. The word ‘positive’ might be recommended.

Reviewer 2 Report

The presented work is of great scientific and practical interest. Hypertriglyceridemia is a form of familial hypercholesterolemia. The authors studied a fairly large group of patients with such a rare pathology. An increase in the level of triglycerides is common in clinical practice and requires a differential diagnostic search for its causes. The work shows how important it is to conduct a genetic study to find the hereditary nature of the disease. An important aspect is the study of the relationship between genetic and non-genetic factors. The interactions found by the authors with triglyceride levels and body mass index are of great importance in the management of such patients.

The paper presents the results in detail. But the unusual construction of
the article caused bewilderment.
In my opinion, the
«Materials and Methods» section should come before
the results and discussion of the data obtained.
It is also noteworthy that in the list of references only two works are dated
2019 and 2020, although the problem of hypertriglyceridemia is now
discussed in great detail in current scientific journals.

Round 2

Reviewer 1 Report

The paper was well revised. Only one point can be reconsidered. In 4.3 Biochemical analysis in Methods (row 279-), …the expression “LDL:” can be changed to “LDL-c:”. After that, the expression “LDL Cholesterol” can be changed to “LDL-c” because the abbreviation already appeared in the earlier text. Similarly, the expression “HDL:” can be changed to “HDL-c:”.
